# Thyroid Hormone Deiodination—Mechanisms and Small Molecule Enzyme Mimics

**DOI:** 10.3390/biom15040529

**Published:** 2025-04-04

**Authors:** Debasish Giri, Govindasamy Mugesh

**Affiliations:** Department of Inorganic and Physical Chemistry, Indian Institute of Science, Bangalore 560012, India; debasishgiri@iisc.ac.in

**Keywords:** enzyme mimetics, iodine, selenium, thyroid hormones

## Abstract

Thyroid hormones, biosynthesized in the follicular cells in the thyroid gland, play a crucial role in regulating various important biological processes. The thyroid hormone is synthesized as pro-hormone L-thyroxine (T4), while the active form is primarily produced through the phenolic ring deiodination of T4 by iodothyronine deiodinase enzymes (DIOs). Three distinct isoforms of the enzyme are known, which, despite having almost similar amino acid sequences in their active site, differ in their regioselectivity of deiodination towards T4 and its metabolites. However, the precise mechanism and the origin of the differences in the regioselectivity of deiodination by DIOs are still not fully understood. Over the years, several research groups have attempted to mimic this system with small molecules to gain some insight into the reactivity and mechanism. In this review, we will explore the recent developments on the biomimetic deiodination of T4 and its derivatives by using selenium-based enzyme mimetics. For example, naphthalene-based molecules, featuring a 1,8-dichalcogen atom, have been shown to perform tyrosyl ring deiodination of T4 and T3, producing rT3 and 3,3′-T2, respectively. The modification of the electron density around the phenolic ring through substitutions in the 4′-hydroxyl group can alter the regioselectivity of the deiodination by deiodinase mimics. Additionally, we will highlight the recent progress in the development of a dipeptide-based DIO1 mimic, as well as the deiodination of other halogenated thyronine derivatives by mimics.

## 1. Introduction

Thyroid hormone production is exclusively carried out by the thyroid gland, which is situated in the lower front part of the neck. It primarily secretes the pro-hormone L-thyroxine (T4) or 3,5,3′,5′-tetraiodo-L-thyronine and a smaller amount of its active form, 3,5,3′- triiodo-L-thyronine (T3). These are the only known iodine-containing biomolecules in the human body and play a vital role in regulating various physiological processes, such as protein synthesis, carbohydrate and fat metabolism, development, growth and maturation, as well as the functioning of the cardiovascular and renal systems [1,2,3]. The synthesis of thyroid hormones is tightly regulated through a feedback loop involving two important organs, the hypothalamus and pituitary gland [4]. The biosynthesis of thyroid hormones occurs in the follicular cells of the thyroid gland, which are rich in the tyrosine-containing protein thyroglobulin (Tg). The heme-containing enzyme thyroid peroxidase (TPO) catalyzes the iodination of tyrosine residues on Tg, producing a mixture of 3-iodotyrosine (MIT) and 3,5-diiodotyrosine (DIT). The phenolic coupling of two DIT molecules in the presence of TPO generates Tg-bound 3,5,3′,5′-tetraiodothyronine (T4). Finally, T4 is released from Tg by proteolysis [1]. Once synthesized, it is transported to target organs by transporter proteins, including thyroxine-binding globulin (TBG), transthyretin (TTR), and human serum albumin (HSA) [1]. However, for the cellular uptake, T4 relies on specific transporters, with monocarboxylate transporter 8 (MCT8) being the most selective and specific for T4 [5]. Other transporters such as MCT10 and organic anion transporter 1c1 (OATP1C1) have also been recently identified and play a role in the cellular uptake of T4 and T3 [6,7]. Although the thyroid gland produces the majority of T4, its active form, T3, is generated through the 5′-deiodination of T4 by a group of enzymes known as iodothyronine deiodinases (DIOs). These enzymes, containing a catalytically important selenocysteine residue in the active site, help convert T4 into the biologically active form, T3 [8,9,10,11]. The action of DIOs is essential for maintaining thyroid hormone homeostasis in the body. T3 has a higher binding affinity (K_d_ = 0.06 nM) for thyroid hormone receptors (TRα and TRβ) compared to that of T4 (K_d_ = 2 nM), which explains its greater biological importance [12]. The lower binding affinity of T4 is attributed to the presence of an additional bulky iodine atom in the phenolic ring, which creates steric hindrance, reducing its ability to bind to thyroid hormone receptors [13]. T3 interacts with nuclear thyroid hormone receptors, binding to thyroid hormone-responsive elements (TREs) in the genes of target cells. This interaction regulates gene expression through the help of co-activators and co-repressors [14,15,16].

In this review, we will explore the recent advancements in the biomimetic deiodination of T4 and its analogues by using synthetic selenium-based compounds. Additionally, we will discuss how the structural modifications in the mimics and T4 can alter reactivity and regioselectivity. Particularly, modifying the halogen bond-forming ability of the iodine atom can alter the regioselectivity of the deiodination of T4 and its analogues. These insights may contribute to the development of novel therapeutic approaches for thyroid hormone-related disorders and improve our understanding of thyroid hormone metabolism.

## 2. Biomimetic Deiodination of Thyroxine

Thyroid hormones, produced by the thyroid gland, are crucial for regulating a variety of physiological functions in the human body. The thyroid primarily produces the prohormone L-thyroxine (T4, or 3,5,3′,5′-tetraiodothyronine), along with small amounts of the biologically active hormone, 3,5,3′-triiodothyronine (T3). However, most of the T3 in the body is generated by a group of enzymes known as DIOs, which contain the catalytically significant selenocysteine residue in the active site. There are three isoforms of DIOs known, which are homologous in nature. They share almost similar amino acid residues at their active sites, but they differ in their activity towards the deiodination of thyroid hormones and their tissue distribution [1].

DIO1 catalyzes the 5′ and 5-deiodination of T4 to produce the biologically active hormone T3 and its inactive metabolite reverse T3 (rT3), respectively. It is primarily expressed in the liver, kidneys, and thyroid. In contrast, DIO2, which is found in the brain, skeletal muscle, and brown adipose tissue, is mainly responsible for producing active T3 through 5′-deiodination. The third isoform, DIO3, specifically catalyzes 5-deiodination and plays a key role in inactivating thyroid hormones when their levels exceed the optimal concentration [17,18,19]. DIO3 is particularly important during fetal development and early life, as it helps protect developing tissues from excessive thyroid hormone activity that could interfere with normal growth and differentiation [20]. The regulation of DIOs expression are crucial as their enzymatic deiodination actions fine-tune the thyroid hormone status in various tissues, ensuring proper metabolic function, growth, and development [21,22,23,24].

Given the importance of regioselective deiodination mediated by iodothyronine deiodinases (DIOs), there has been increasing interest in developing selenium-based small molecules as mimetics of these enzymes. Engman and co-workers demonstrated the deiodination of 2,6-diiodophenol derivatives by using selenium and tellurium-based reagents such as PhSeH, PhTeH, and NaHTe. Similarly, it was observed that the tellurium reagents (NaHTe and Na_2_Te) can remove the iodine atoms from the phenolic ring of an N-butyryl methyl ester derivative of T4, while the iodine atoms on the tyrosyl ring remain unaffected [25]. In contrast, the corresponding selenium reagents were unable to achieve this conversion [25]. Later, Goto et al. for the first time reported that a sterically hindered selenol-based compound capable of mediating the 5′-deiodination of N-butyryl-thyroxine methyl ester in the presence of triethylamine (Et_3_N), similar to the process observed with DIO2. The deiodination reaction took place in CDCl_3_ as the solvent, although it proceeded at a slower rate (7 days at 50 °C). This study demonstrated the formation of a selenenyl iodine intermediate as proposed earlier for DIO1. A keto-enol mechanism was proposed for 5′-deiodination, suggesting that the compound was inefficient in removing the 5-iodine atom from the thyroxine derivative [26].

Later, studies revealed that halogen bonds (XBs) may play a significant role in the deiodination of thyroxine. In organic halides, the electron density around the halogen atom is distributed anisotropically, creating a positive charge (σ-hole) along the C-X (X = Cl, Br, I) axis [27,28,29]. This σ-hole facilitates non-covalent interactions between the halogen and an electron donor, forming highly directional halogen bonds (XBs) [30]. Halogen bonding plays crucial roles in various applications, ranging from drug design to the cellular delivery of small molecules [31,32,33,34,35,36]. This can be better understood through molecular orbital theory, where the electron density from the lone pair of a donor atom (D) is donated to the antibonding σ* orbital of the accepting halogen (A) atom. The strength of these interactions (ΔE_D→A_) is inversely proportional to the strength of the C-X bond; a weaker C−X bond leads to greater accessibility of the antibonding σ* orbital for interaction with the donor [30,37]. Halogen bonds have been observed in thyroxine-binding proteins (TBG and TTR) and in thyroid receptors [38,39,40]. More recent study suggests that it plays a crucial role in the recognition of thyroxine and its derivatives by monocarboxylate transporter 8 (MCT8) [41]. Bayse et al. reported that a free selenol can form such XB interactions with iodine atoms in diiodo phenol analogues, with this interaction becoming stronger when the selenol is converted into selenoate [42]. Since the pKa of selenol is lower than the physiological pH, it is expected that it predominantly exists in its selenoate form, which is more nucleophilic in nature. In contrast, the corresponding sulfur analogue forms very weak halogen bond interaction [42]. Theoretical studies suggest that the iodine atom in thyroxine contains σ-hole, which can form such a non-covalent interaction with a free selenol or selenoate moiety. However, a mimic with only a selenol moiety was found to be ineffective in selectively removing the iodine atom from tyrosyl rings under physiological conditions. These observations suggest that additional structural features are required for the deiodination reactions.

It was noted that DIO1 and DIO3 contain highly conserved selenocysteine (Sec126 and Sec170) and cysteine (Cys124 and Cys168) residues in their active sites, which are essential for TH deiodination (Figure 1A). Mutations in either of the residues lead to a loss of deiodination activity [43]. Inspired by this close proximity of selenocysteine and cysteine residues, our group was the first to demonstrate that 1,8-peri-substituted naphthyl-based compounds can mimic DIO3 activity under physiological conditions (Figure 1B). Compound **1**, featuring a thiol–selenol pair at the 1,8-position of naphthalene, selectively mediates the 5-deiodination of T4 and T3, producing rT3 and 3,3′-T2, respectively (Figure 1C) [44]. Replacing the selenol group with a thiol group (**2**) in compound **1** decreased the reactivity, highlighting the importance of the selenol moiety in DIO activity. On the other hand, replacing the thiol with another selenol (as in compound **3**) significantly enhanced the deiodinase activity [45].

The previously observed keto-enol tautomerism-based mechanism in the phenolic ring deiodination was not feasible in this case [26]. Halogen bond-mediated deiodination appears to be involved in the tyrosyl ring deiodination of T4. Theoretical studies indicate that the strength of the σ-hole on the iodine atom present in the tyrosyl ring is stronger than that on the phenolic ring (Figure 2A). Additionally, it was observed that the LUMO, which has a C−I* character, lies in the tyrosyl ring, suggesting that it can form stronger interactions with an electron donor compared to the iodine atoms present on the phenolic ring. This observation was further supported by the halogen bond interaction energy with methyl selenoate, which serves as a mimic for the selenocysteine residue in DIOs. However, halogen bonding alone might not be sufficient to remove the iodine atom. The formation of a Se···I halogen bond (XB) makes selenium in the mimic electron-deficient, which allows the nearby thiol group (compound **1**) to interact with the electron deficient selenium through a Se···S chalcogen bond (ChB) [46]. This cooperative halogen and chalcogen bond interaction leads to the polarization and cleavage of the C-I bond (Figure 2B). Replacing the thiol group with selenium enhances the strength of the chalcogen bond, which in turn strengthens the Se···I halogen bond and facilitates more efficient iodine atom abstraction [46]. Without either of these non-covalent interactions, the mimic becomes inactive, underlining the importance of their role. In the case of DIO3, the selenol and thiol groups are positioned approximately 10 Å apart. However, in the mimic, the rigidity of the aromatic ring brings the selenol and thiol moieties into close proximity, thereby facilitating the removal of the iodine atom under physiological conditions. Introducing a basic amino group (**4**) nearby further enhances the reaction rate without altering the selectivity of the deiodination (Figure 1B). Sicilia et al. validated the reaction mechanism using DFT calculations and suggested that the imidazole moiety of His residues present at close proximity to the selenocysteine at the active site may serve as a general base catalyst, facilitating the removal of protons from the selenol group [47,48]. The use of a naphthyl-based mimic has also been expanded to dehalogenate halogenated nucleosides (Br, I) through halogen bonding [49]. These compounds are known to be harmful, as they can function as carcinogens or DNA photosensitizers, potentially leading to cellular damage or mutations [50]. This halogen and chalcogen bond-based mechanism for deiodination was further supported when selenium atom was replaced with the more polarizable and stronger electron-donating element tellurium. The selenium to tellurium substitution not only increases reactivity but also alters regioselectivity. When compound **6**, having two tellurium atoms, was tested as a mimic, it facilitated the stepwise removal of all four iodine atoms from T4 to produce T0. The deiodination rate with compound **6** was significantly higher than that of compounds **1**–**4** due to the formation of a stronger halogen bond with the tellurium atom of the mimic and the iodine atom of thyroxine (Figure 2C) [51].

While the crystal structure and probable mechanism of DIO3 have been investigated to some extent [43,46,52], the molecular mechanisms of DIO1 and DIO2 remain poorly understood. Small-molecule model compounds that specifically mimic the functions of DIO1 and DIO2 under physiological conditions may help in understanding the molecular mechanism. Recently, the crystal structure of the catalytic domain of mouse DIO2 was solved, which reveals a structure strikingly similar to that of DIO3 [53]. This similarity suggests that both enzymes may share a common mechanism for deiodination. However, to date, no small molecule-based DIO2 mimic has been reported. In a significant development, Arai et al. developed the first peptide-based diselenide molecule that mimics DIO1 activity [54]. Compound **7** featuring a six-membered diselenide ring and a histidine residue connected through a (S)-azetidine-2-carboxylic acid (AZA) bridge, shows deiodination activity in the presence of excess DTT. Upon reduction with DTT, the ring-opened states form a γ-turn (**9**), positioning the selenium atoms near the imidazole ring, as confirmed by one- and two-dimensional NMR and circular dichroism (CD) spectroscopy (Figure 3A) [55]. Typically, the reductive cleavage of a diselenide bond (Se-Se) by a thiol compound is thermodynamically unfavorable. However, in compound **7**, a hydrogen bond between the His side chain and the selenol group stabilizes the formation of the diselenol. The histidine residue further abstracts a proton from the selenol to convert it into a more nucleophilic selenoate, which is capable of forming stronger halogen bonds compared to the selenol group (Figure 3A) [54].

In its reduced state, this compound efficiently removes iodine from both the tyrosyl and phenolic rings of T4 at similar rates, generating rT3 and T3, respectively (Figure 3B). Further deiodination of rT3 and T3 leads to the production of 3,3′-T2, which does not undergo further deiodination. It can also remove 5′ and 5-iodine atoms, respectively, from 3′,5′-T2 and 3,5-T2 to produce the final mono-iodinated products. A similar behavior of deiodination was observed when compound **10**, having a proline spacer between histidine and diselenide, was used. It promoted the reaction at a faster rate than compound **9**. This increased reaction rate is attributed to the more tightly packed γ-turn structure of **10** compared to **9** (Figure 3A) [54,55].

Interestingly, when the strain spacer was replaced with Gly, however, the deiodination activity was unexpectedly abolished, suggesting that the formation of the γ-turn is crucial for the observed deiodination activity. Additionally, replacing His, which forms a salt bridge interaction with the selenol, with alanine (Ala) drastically slowed down the reaction, reinforcing the importance of His in the deiodination process. Interestingly, replacing the diselenide moiety with a seleno-sulfide ring (compound **11**) results in the loss of DIO1-like activity. This is in contrast to the naphthalene-based seleno-sulfide mimic (compound **1**), which exhibits deiodinase activity, although the activity was found to be lower than that of **3**. The loss of activity for compound **11** is attributed to the weaker halogen bond formed between the free SH group in its reduced state and iodine in T4 (conformation II). In another conformation, the steric hindrance of the free SH group prevents T4 from approaching the Se group, thereby hindering the formation of the Se···I intermediate (Figure 3C). These observations suggest that in DIO1, in addition to the halogen bond formation between the selenium and iodine atoms, the specific conformation change after binding with T4 may also play a crucial role in its regioselective deiodination [54].

## 3. Crystal Structure of DIOs and Role of Conserved Cys Residue

Following our studies on the biomimetic inner-ring deiodination of thyroid hormones (THs), the crystal structure of DIO3 was reported by Schweizer et al. (Figure 4A) where the Sec170 residue was replaced by Cys170 in their construct [43]. The Sec170 is positioned in the loop connecting β1 to α1, resembling the location of the peroxidatic cysteine found in thiol reductases such as peroxiredoxins (Prx) and glutathione peroxidases (GPX), among others. Multi-microsecond molecular dynamics (MD) simulations of the DIO3 crystal structure revealed the formation of a narrow cryptic pocket around catalytically active Sec170, which is observed in one of the conformations based on the electrostatic potential of DIO3^trunc^ (Figure 4C) [56]. In this conformation, Sec170 is partially buried within the pocket, shielding it from cellular oxidants. Phe258 creates a hydrophobic wall, supported by Tyr257 and Ala201 near Sec170. This pocket accommodates T4, which is facilitated by an inner ring interaction involving the TH-Sec170 I···Se halogen bond (XB). Additionally, an I···O halogen bond is formed between the tyrosyl ring iodine of T4 and the Asp211 carboxylate group, with distances ranging from 3.0 to 3.5 Å and a C–I···O bond angle between 165° and 180°, indicative of a strong halogen bond interaction (Figure 4C) [56]. The other phenolic ring iodine remains in a hydrophobic region created by the sidechains of Ala201 and Phe258. This secondary I···O XB interaction with Asp211 contributes to the high affinity of DIO3 for T3, as compared to other substrates without phenolic ring iodine atoms such as the one in 3,5-T2. By superimposing the T3 binding from the His435-Arg282 clamp of the T3/T3Rβ complex onto DIO3-His202-Arg275, a preliminary DIO3-T3 substrate complex model is generated (Figure 4D) [43,52]. Triiodothyronine is present in the DIO3 substrate-binding cleft with the 5-iodine atom approximately 3–4 Å away from the Se center of Sec170. The His202 in DIO3 serves as a binding partner for the 4′-phenolic group of iodothyronine, suggesting that the His202 residue probably does not play a direct role in the catalysis, as previously reported. On the other hand, the interaction between the carboxyl group of the substrate and the guanidinium group of Arg275 plays a crucial role in 5-deiodinase activity. A sharp reduction in the activity was observed for the Arg275Ala mutant. Additionally, Glu259 positioned close to Arg275 may assist in the recognition of the iodothyronine amino group (Figure 4D).

The conserved cysteine residues in DIO1 (Cys124 and Cys194) and DIO3 (Cys168 and Cys239), located near the active selenocysteine residue, may function as resolving cysteines in a Prx-like deiodination mechanism. For example, in 2-Cys peroxiredoxins (Prx), the first catalytic half-reaction results in the formation of a reduced product and an intramolecular disulfide bond between the peroxidatic cysteine and a resolving cysteine [57]. This disulfide is subsequently reduced by thioredoxin (Trx). Similarly, in DIO3, the initial formation of a selenenyl iodide intermediate may be formed when the iodine is abstracted from the 5-position of T4 through a halogen bond interaction. This intermediate can rapidly react with either Cys168 or Cys239, forming a selenenyl sulfide bond. However, it was observed that Cys168 is closer to the active site than Cys239 as Cys168 only requires minimal rearrangement to attack the selenenyl iodide bond. Mutation studies also indicate lower enzymatic activity for the Cys168Ala mutant, while no change is observed for the Cys239Ala mutant, suggesting that Cys168 is more important for the enzymatic reaction [43]. After the initial formation of the Sec170-Cys168 bond, the proximal Cys239 properly reorients itself for a nucleophilic attack at the Sec170-Cys168 bond, resulting in the formation of Cys168-Cys239 disulfide. This disulfide bond is more exposed on the surface, making it more accessible for the reduction. Biochemical studies further support that this oxidized form can be readily reduced by external thiols. This proposed model further suggests that proton transfer to the 5-position of the iodothyronine occurs through a network involving His219, Glu200, and Ser167, with Tyr197 and Thr169 participating in a hydrogen bond network. Mutational studies have supported the crucial role of this hydrogen bond network in the catalytic process [43,52]. In the case of DIO2, the active site cysteine (Cys128 in mouse and Cys131 in human) is replaced by an Ala128 residue, whereas the Cys205 residue (Cys239 in DIO3) is conserved (Figure 4B), and it acts as a single-turnover enzyme and is more susceptible to ubiquitin tagging [53,58].

## 4. Biomimetic Deiodination of Thyroxine Derivatives

It was observed that the naphthyl-based deiodinase mimics can mediate the tyrosyl ring deiodination of the decarboxylated thyroid hormone metabolite thyronamine, which is known to be a substrate for DIOs [59]. However, this deiodination occurs at a slower rate. The observed decrease in deiodination rate can be attributed to the weaker halogen bond interaction between the iodine and selenium atoms compared to that observed in T4. Recently, we have found that the regioselectivity of the biomimetic deiodination of T4 by naphthyl-based mimics can be modulated by introducing electron-donating or electron-withdrawing groups at the 4′-O position of T4. For example, attaching a -SO_3_H group to the hydroxyl group not only accelerates the rate of deiodination but also facilitates both 5- and 5′-deiodination by deiodinase mimics [60]. Similarly, replacing the –OH group with other electron-donating groups such as -OMe, -OEt, or -OPr^i^ promotes both 5- and 5′-deiodination by compound **3** (Figure 5A). Interestingly, introducing an electron-withdrawing group like -OCH_2_CN (**13**) completely shifts the regioselectivity from the tyrosyl ring to the phenolic ring (Figure 5A). This effect is further enhanced when a strong electron-withdrawing -OCF_2_H group (**14**) is introduced, which not only accelerates the 5′-deiodination but also facilitates the removal of all four iodine atoms, resulting in the generation of the corresponding T0 analogue (**20**), similar to the observation made for the tellurium-based mimics (**5**,**6**) (Figure 5B,C) [50,61].

An earlier NBO analysis showed that the 3- and 5-iodine atoms in T4 have a more positive charge (0.208) than the 3′- and 5′-iodines (0.191), indicating that the 3,5-iodines are more prone to form stronger halogen bonds with electron donors like selenium (Figure 6A). This explains why T4 undergoes exclusive 5-deiodination under physiologically relevant conditions by the mimics. Substituting the -OH group in T4 with a methoxy group has little effect on the charge, while introducing an electron-withdrawing group like -OCH_2_CN increases the positive potential on the phenolic ring iodine to 0.206, and further to 0.214 with the -OCF_2_H group (Figure 6A) [61]. This is supported by the deeper σ-hole formation in the phenolic ring iodine in compound **14** (Figure 6B). Halogen bond strength calculations using methyl selenoate show that in T4, the tyrosyl ring iodine has a higher interaction energy (57.34 kcal/mol) than the phenolic ring (49.22 kcal/mol). However, with the -OCF_2_H substitution, the halogen bond (XB) energy involving a phenolic ring iodine is increased by 16 kcal/mol, surpassing the tyrosyl ring’s energy (Figure 6C). The reversal of the strength of interaction energy elongates the C-I bond in the phenolic ring more, demonstrating that the electron-withdrawing substitutions significantly alter the electronic properties of the iodine atoms and drive the complete reversal of regioselectivity of the deiodination mediated by the synthetic mimics.

As the inclusion of an electron-withdrawing group at the 4′-OH position enhances phenolic ring deiodination by strengthening the halogen bond, we further investigated the effect of electron-withdrawing substitutions by replacing the iodine atom with more electron-withdrawing halogen atoms (F, Cl, and Br) on both tyrosyl and phenolic rings [62]. When the fluoro-analogue was used as a substrate, it underwent tyrosyl ring deiodination exclusively, similar to T4, and this pattern was observed in all other compounds (Figure 7A). As halogen bonding plays a crucial role in the deiodination, we found that Cl and Br can form reasonably strong halogen bonds with Lewis bases, much like iodine. However, no dechlorination or debromination was observed during the biomimetic reactions with the naphthyl-based mimics (Figure 7A). DFT calculations showed that the substitution with more electronegative halogen atoms did not significantly increase the charge on the iodine atom.

The electron-withdrawing effects of F, Cl, or Br on the phenolic ring are likely to be compensated by the electron-donating effect of the hydroxyl group. Similarly, for all other compounds, the charge on the iodine atom attached to the tyrosyl ring was more positive than that on the phenolic ring (Figure 7B). To understand the inability of the mimic to mediate this dechlorination or debromination, we examined the frontier molecular orbitals (FMOs) with C-X* characters. Halogen bonding is best described as the donation of electron density from the donor to the anti-bonding orbital of the C-X bond. As with T4, the C-I* character FMOs are mostly localized in the LUMO or LUMO + 1 orbitals, whereas the molecular orbitals with C-Cl* and C-Br* characters are delocalized into higher-energy orbitals. For example, in compounds **21** and **22**, the C-Cl* and C-Br* character molecular orbitals are located in LUMO + 10 and LUMO + 7, respectively (Figure 7C). Halogen bond interaction energy calculations with a model selenium compound suggest that the Cl⋅⋅⋅Se (5–7 kcal/mol) and Br⋅⋅⋅Se (26–32 kcal/mol) interactions are much weaker than the I⋅⋅⋅Se (48–57 kcal/mol) interactions. A threshold energy of more than 50 kcal/mol is required for the activation of the C-X bond present in different thyroxine derivatives. The lack of dechlorination or debromination in compounds by the deiodinase mimic can be attributed to the significantly lower positive charges on chlorine (0.037) and bromine (0.103), the high-energy anti-bonding molecular orbitals, and the weak halogen bonding between the halogens and selenium atom [62]. This is consistent with the previous findings that polybrominated diphenyl ethers (PBDEs) and polychlorinated biphenyls (PCBs) act as inhibitors rather than substrates for dehalogenation [63,64]. PBDEs and PCBs are likely to alter thyroid hormone levels by inhibiting deiodinases (DIOs), thereby acting as endocrine-disrupting agents. In model studies, halogen bond interaction energies were calculated by using MeSe- as the halogen bond acceptor. It was observed that the halogen bond energy follows the order THs > PBDEs > PCBs [65]. In the case of THs, the halogen bond energy is sufficient to facilitate the removal of iodine atom by DIOs, except for 3-T1, which does not undergo deiodination. Based on these observations, a threshold energy (energy of donor + acceptor complex formation (ΔE + ZPE)) lower than −21.42 kcal/mol (ΔE + ZPE = −21.42 kcal/mol for 3-T1) was proposed as a requirement for any dehalogenation reaction to occur with DIOs [66]. Although most PBDEs have a higher zero-point energy than THs, some PBDEs, particularly highly substituted PBDEs and OH-BDEs, can undergo debromination by DIOs, thereby inhibiting the TH deiodination activity [63]. In contrast, compounds with chlorine substitutions form very weak halogen bonds, and therefore, they may not undergo any dechlorination by DIOs. As a result, PCBs can inhibit DIO activity by binding reversibly with the DIOs [64]. Although halogen bond interactions play a crucial role in the inhibition of DIOs by these endocrine-disrupting agents, factors such as the binding of the hydroxyl group to the active site, the conformation of the biphenyl moiety, and the position of the halogen atom in the ring also play important roles in their inhibitory effects [66].

## 5. Potential Role of Thyroxine Conformations in Deiodination

Although the deiodination mechanism and crystal structure of some of the DIOs are well established, the regioselectivity of deiodination by DIOs remains unclear. While DIO1 and DIO3 both contain conserved cysteine and selenocysteine residues in their active sites, the naphthyl-based mimic with a thiol and selenol in close proximity to each other was unable to remove the phenolic ring iodine from T4. It should be noted that the iodine atom is removed from the tyrosyl ring through co-operative halogen and chalcogen bond interactions by the naphthyl-based mimetics. While the halogen bond is widely accepted to play a crucial role in the activation of the C-I bonds by DIOs, there is no evidence for the formation of a chalcogen bond interaction in DIO3, as the thiol moiety is positioned far from the active selenocysteine residue. Furthermore, in the deiodination process by DIOs, the interactions between the substrate and side chain amino acid residues along with changes in the conformation of the active site during substrate binding may play a key role, whereas such interactions are absent in biomimetic reactions. This suggests that other factors are likely to influence the reactivity and regioselectivity of the deiodination of T4 by DIOs. Such factors may also alter the strength of the halogen bond.

Our group has made significant effort to understand the differences in their regioselectivity. They found that T4 exists in various polymorphic forms in the solid state, each displaying distinct solubility, optical activity, and spectroscopic characteristics [67]. Similarly, it was observed that T4 can bind to its relevant proteins, such as TBG, TTR, etc., with different binding modes (Figure 8A,B) [68]. They bind not only in either the cisiod or transiod conformations but also, even within the same conformation, they exhibit different structural parameters (Figure 8A). Interestingly, when the halogen bond energy was calculated using methyl selenoate as the halogen bond acceptor, the interaction energies for the phenolic and tyrosyl ring iodine atoms varied significantly for different conformations of T4. In most conformations, the iodine atom on the tyrosyl ring forms a stronger XB than the iodine on the phenolic ring. However, for certain conformations, such as when the φ and φ’ values are 0° and 94.8°, respectively, the XB energy for the phenolic-ring iodine becomes comparable to that of the tyrosyl-ring iodine (Figure 8C). For this orientation, the molecule may show both 5 and 5′-deiodination. Furthermore, adjusting the orientation of the amino acid moiety (determined by χ1 and χ2) at this φ value (0°) resulted in the XB energy for the phenolic-ring iodine being stronger than that for the tyrosyl-ring iodine (Figure 8D) [67]. For this orientation, the molecule may exhibit only 5′-deiodination. These findings suggest that DIOs may influence the regioselectivity between tyrosyl- and phenolic-ring deiodination by inducing conformational changes in T4 at their active sites [67,69].

Furthermore, we have observed that the histidine residue may play a key role in determining the regioselective deiodination of T4 by DIOs. The histidine at position 202 in DIO3 is also conserved in DIO1 (position 158) and DIO2 (position 162). The mutation of His158 in DIO1 leads to a complete loss of enzyme activity [70]. From the crystal structure of DIO3-T3, it is observed that His202 forms a hydrogen bond with the hydroxyl group of T3, which is similarly observed in DIO2, where the hydroxyl group of T4 interacts with the His 162 residue [43,53]. Such binding may influence the halogen bond-forming ability of the iodine atom specifically on the phenolic ring. When we performed the DFT calculations of the T4-Im complex (where imidazole mimics histidine), we observed that T4 can act either as a hydrogen bond donor (HBD) or acceptor (HBA) (Figure 9). When T4 acts as an HBD through its hydroxyl group, the charges on the phenolic ring iodine atoms decreased significantly as compared to that of T4 alone [61]. On the other hand, when T4 acts as an HBA, there is a slight increase in the charge on one of the iodine atoms (0.185 compared to 0.174 for T4) (Figure 9). Previous simulation studies with the DIO3-T4 complex showed that protonated His202 is positioned between the outer ring of T4 and Sec170, and deprotonation of the histidine residue results in the loss of T4 binding [56]. As histidine can exist in different forms depending on pH and the positively charged histidine is present in various proteins, we also examined the effect of a protonated imidazole (ImH). In this case, imidazole exclusively acts as an HBD, and the charges on the phenolic ring iodine atom are increased significantly (Figure 9). As we know, an increase in the positive potential strengthens the halogen bond with selenoate. Therefore, one might expect that hydrogen bonding with a protonated imidazole could favor 5′-deiodination, as observed in DIO1 and DIO2 [1].

## 6. Conclusions and Future Perspectives

Thyroid hormones (THs) are essential for nearly every cell in the human body as they regulate various physiological functions, including metabolism, metabolic rate, protein synthesis, bone growth, neuronal development, and cardiovascular health. Although the thyroid gland primarily produces the prohormone L-thyroxine (T4), its biologically active form T3 is generated through deiodination, a process involving the rare amino acid selenocysteine. Iodine plays a crucial role in activating the hormone, as well as in the function of its transporters and membrane transport. The formation of halogen bonds appears to play several key roles in the binding of the hormone to its receptors and transporters, and in the activation/deactivation processes. Although halogen bond-mediated deiodination by DIOs is generally accepted, the regioselectivity of the deiodination by these enzymes remains a mystery. The binding of T4 in different conformations as well as the potential interactions with side-chain amino acid residues like His may influence the regioselectivity of the deiodination reactions. Future research in this area may focus on gaining a deeper understanding of the mechanism of regioselective deiodinations by DIOs, based on their differences in the crystal structure as well as binding within active sites. While mimics for both DIO1 and DIO3 have been reported, there have been no reports on a DIO2 mimic which is capable of removing iodine exclusively from the phenolic ring of T4 to produce T3 under physiological conditions. Modulating the activity of deiodinases (DIOs) holds significant potential as a therapeutic strategy for treating various thyroid disorders. A deeper understanding of the molecular mechanisms underlying deiodinase function could aid in the development of specific inhibitors for these enzymes. For example, by using a DIO3 mimic, we could identify small molecules that specifically target the DIO3 enzyme, offering a novel treatment strategy for ovarian cancer, which depends on DIO3 activity. Finally, the identification of cofactor(s) which can regenerate the active site after a cycle of deiodination is crucial for understanding the catalytic nature of the deiodinases.

## Figures and Tables

**Figure 1 biomolecules-15-00529-f001:**
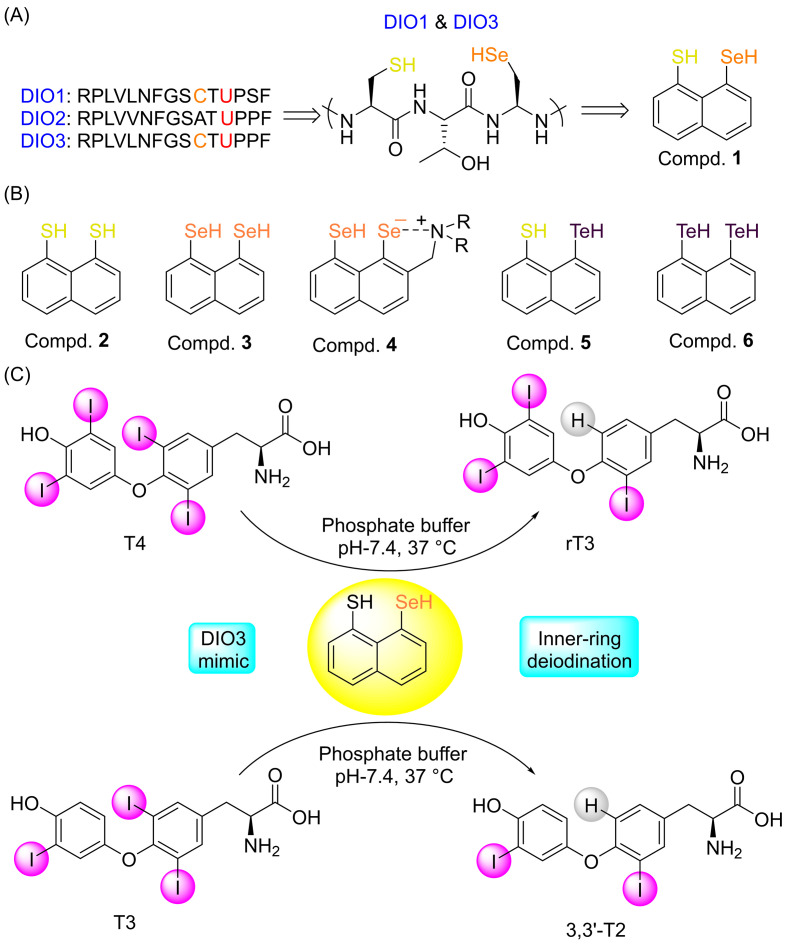
(**A**) Amino acid sequences surrounding the selenocysteine residue in iodothyronine deiodinases (DIOs), showing they are homologous in nature in the region of active site. Additionally, cysteine and selenocysteine are located in close proximity to each other in DIO1 and DIO3. (**B**) Designing mimics (**1**–**6**) with 1,8-dichalcogen substituted naphthalene-based compounds. (**C**) Tyrosyl ring deiodination of T4 and T3 to produce rT3 and 3,3′-T2, respectively, using the mimic under physiological conditions. Modified from reference [44].

**Figure 2 biomolecules-15-00529-f002:**
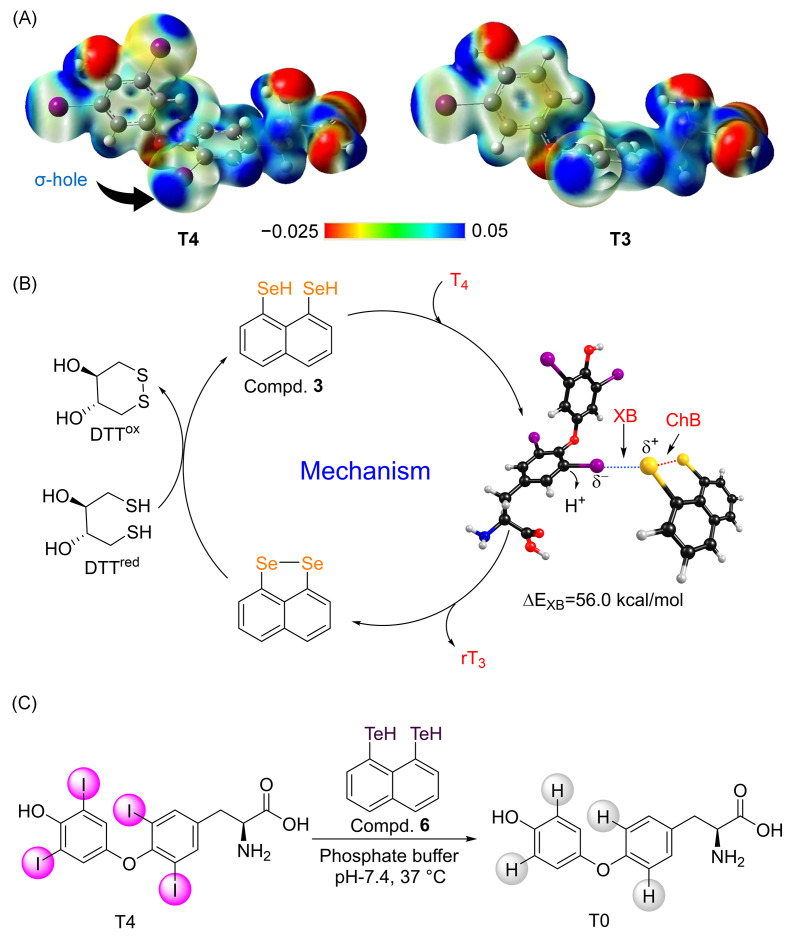
(**A**) ESP diagram of T4 and T3 showing the presence of a sigma hole (σ) on the iodine atoms. (**B**) Mechanism of deiodination of T4 by compound **3**, showing that both halogen and chalcogen bonds are responsible for the selective tyrosyl ring deiodination. (**C**) Deiodination of T4 to T0 by a tellurium atom containing mimic **6**. Adapted from reference [46].

**Figure 3 biomolecules-15-00529-f003:**
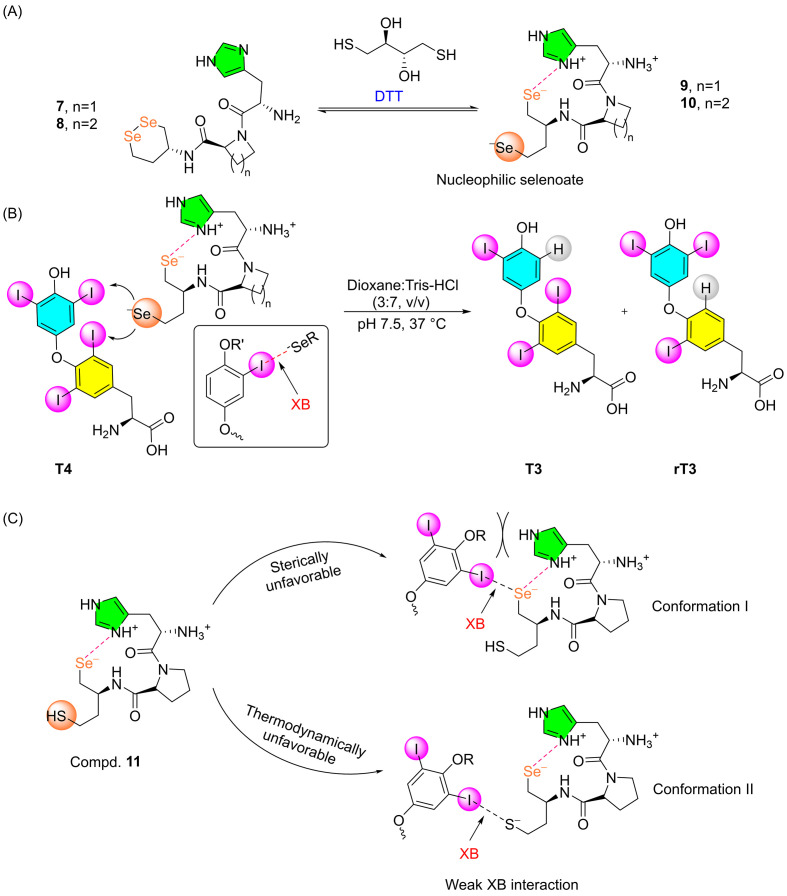
(**A**) Structure of cyclic diselenides conjugated with His through a spacer (Aze or Pro) and corresponding to their open-chain derivatives. (**B**) 5 and 5′-deiodination of T4 by compound **9** and **10** under physiological condition. (**C**) Prohibited halogen bond formation with compounds **11**. Adapted from reference [54].

**Figure 4 biomolecules-15-00529-f004:**
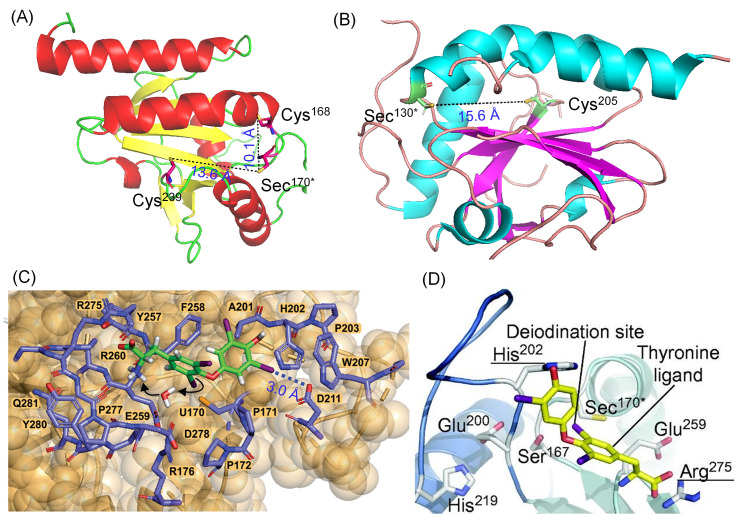
(**A**,**B**) Crystal structure of the mouse DIO3 (PDB Code:4TR3) and DIO2 (PDB Code:9H48) catalytic domain. Sec170* and Sec130* indicates that Sec170 and Sec130 are replaced by Cys170 and Cys130 in DIO3 and DIO2, respectively in their construct. (**C**) Thyroxine binding to the cryptic pocket of DIO3. Reproduced from reference [56]. (**D**) Modelled T3 binding with DIO3 in the active site. Reproduced with permission from reference [43].

**Figure 5 biomolecules-15-00529-f005:**
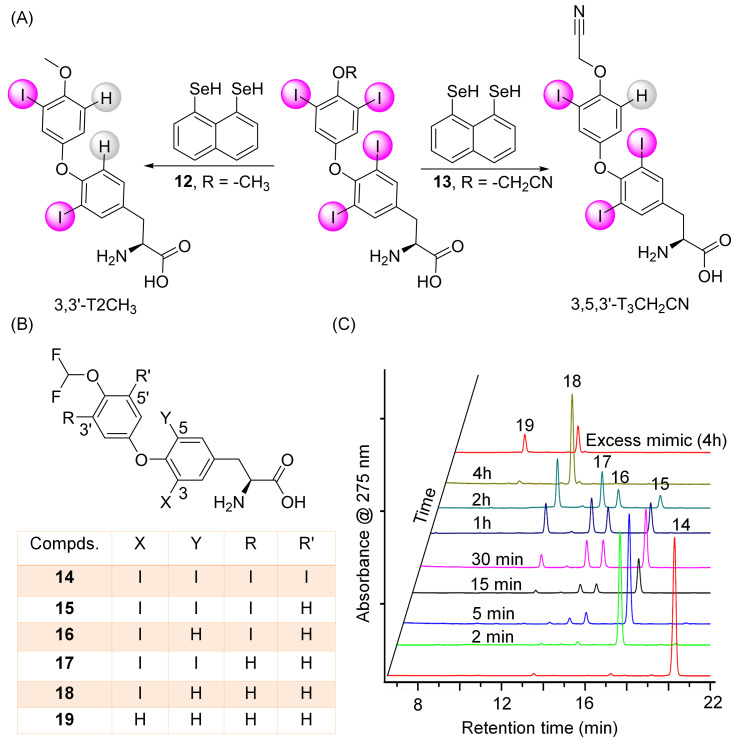
(**A**) Deiodination of compounds **12** and **13** by mimic 3. (**B**) Chemical structures of compounds **14** and its derivatives. (**C**) HPLC chromatogram of the deiodination of compound **14** with mimic **3** showing the formation of different derivatives. Modified from reference [61].

**Figure 6 biomolecules-15-00529-f006:**
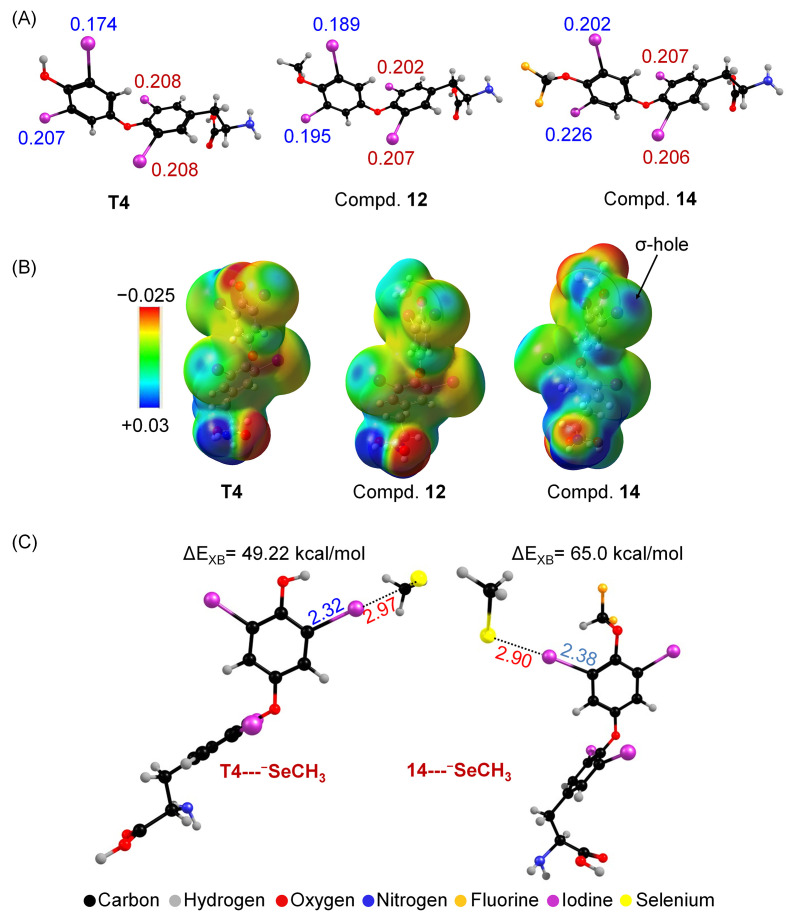
(**A**) The Natural Population Analysis (NPA) charges indicated for the iodine atoms were calculated using an NBO analysis. (**B**) Electrostatic potential map of compounds T4, **12**, and **14**. (**C**) Optimized structures of compounds T4 and **14** with methyl selenoate, showing the formation of the halogen bond interaction. All the distances are given in Å.

**Figure 7 biomolecules-15-00529-f007:**
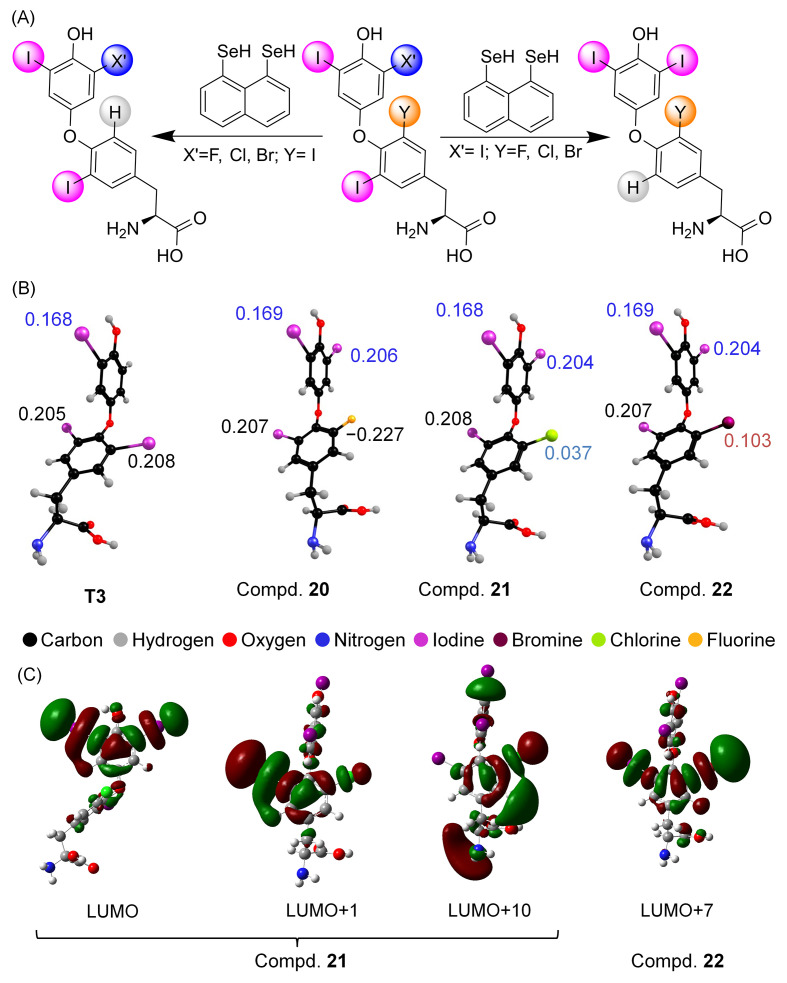
(**A**) Deiodination of the halogen analogues of T4, where one of the iodine atoms is replaced with another halogen atoms (F, Cl, or Br) in each ring. (**B**) The Natural Population Analysis (NPA) charges indicated for the halogen atoms for T3 and **20**–**22** were calculated using an NBO analysis. (**C**) LUMO and next LUMOs for compounds **21** and **22** with a C-X* character. Modified from reference [62].

**Figure 8 biomolecules-15-00529-f008:**
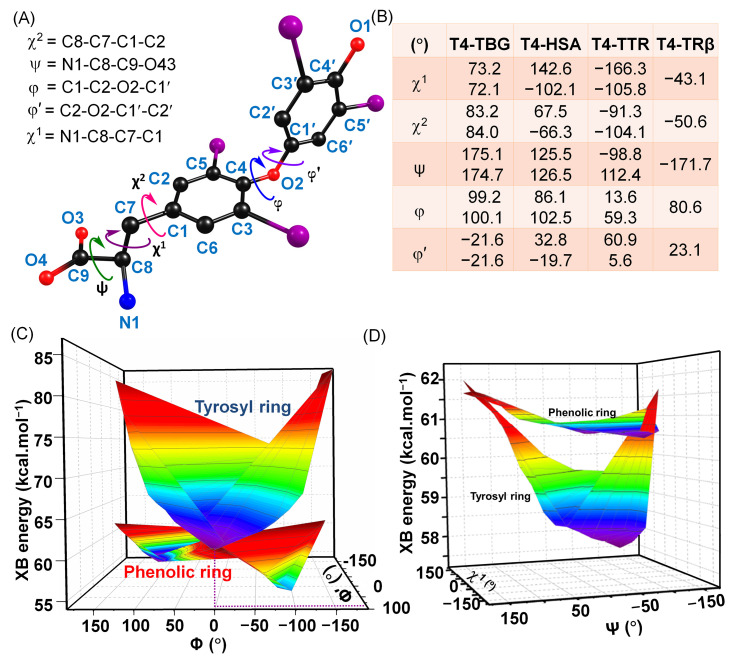
(**A**) Different conformational parameters of T4. (**B**) Different parameters observed in different T4-protein complexes. Adapted from reference [67]. (**C**) Effects of Φ and Φ′, and (**D**) χ1 and ψ at Φ = 0° on the strength of XBs formed between 5- and 5′-iodine of T4 and methyl selenoate. Reproduced, with permission, from reference [67]. Copyright 2015, John Wiley and Sons.

**Figure 9 biomolecules-15-00529-f009:**
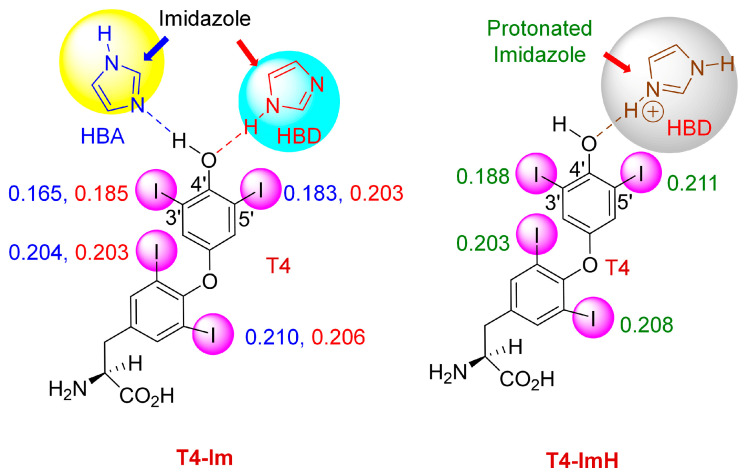
Imidazole complex showing two different modes of hydrogen bonding of T4 with imidazole that acts as either a hydrogen bond acceptor (HBA) or a hydrogen bond donor (HBD). Adapted from reference [61].

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
