# Peer review of "Thyroid Hormone Deiodination—Mechanisms and Small Molecule Enzyme Mimics"

_biomolecules, 2025, doi:10.3390/biom15040529_

Round 1

Reviewer 1 Report

Comments and Suggestions for Authors

This is a nice review of Mugesh and others' work on mimics of the iodothyronine deiodinase family of selenoproteins. It can be published with minor revisions below:

Fig 4 - PDB 4TR3 is Dio3, not Dio1

Pg 13, lines 368-385 - I am in agreement with the discussion of the donation into the C-X* antibonding bond, but note that the ideas presented heavily overlap with refs 63 and 64, which are added in brief without tying into the concepts of the rest of the paragraph, in particular the idea of a threshhold for dehalogenation, which is discussed in ref 64, but also in a subsequent paper (Figure 9 in mdpi.com/1420-3049/25/6/1328, or paper sciencedirect.com/science/article/pii/S2452223621000419), but not attributed. Tying the results on Cl,Br-substituted molecules with this previous work, not in detail, but briefly, would strengthen the importance of the authors' conclusions.

Author Response

Thank you very much for taking the time to review this manuscript. Please find below the detailed responses along with the corresponding revisions and corrections, which are highlighted in track changes in the re-submitted files.

Reviewer comment 1: Fig 4 - PDB 4TR3 is Dio3, not Dio1

Our response: We have updated the name of DIOs code for the corrected PDB code.

Reviewer comment 2: Pg 13, lines 368-385 - I am in agreement with the discussion of the donation into the C-X* antibonding bond, but note that the ideas presented heavily overlap with refs 63 and 64, which are added in brief without tying into the concepts of the rest of the paragraph, in particular the idea of a thresh hold for dehalogenation, which is discussed in ref 64, but also in a subsequent paper (Figure 9 in mdpi.com/1420-  3049 /25/6/1328, or paper sciencedirect.com/ science/article/ pii/S2452223621000419) but not attributed. Tying the results on Cl,Br-substituted molecules with this previous work, not in detail, but briefly, would strengthen the importance of the authors' conclusions.

Our response: As suggested by the reviewer, we have further expanded on the halogen bond interactions between the polybrominated diphenyl ethers (PBDEs) and polychlorinated biphenyls (PCBs) and methyl selenoate in the context of their potential role in the DIOs inhibitors. This is highlighted in the revised manuscript. We have also included additional references to for the detailed discussion (Ref. no-65,66).

Reviewer 2 Report

Comments and Suggestions for Authors

This review article provides an insightful overview of the recent advances in the biomimetic deiodination of T4 and its analogues using synthetic selenium-based compounds. The authors primarily discuss their investigations with naphthalene-based mimics, while also covering studies employing dipeptide-based DIO1 mimics. In addition, the review includes theoretical analyses regarding the potential role of thyroxine conformation and the potential interactions with side-chain amino acid residues such as histidine during deiodination. Overall, the article offers highly valuable information for researchers in this field, and I believe it is suitable for publication in Biomolecules provided that the following points are addressed:

1. On lines 175–177, the authors state,  

  “Due to the rigidity of the aromatic ring, the selenol and thiol moieties are held close to each other, closely mimicking the primary structure of DIO3.”

  However, as noted in lines 276–278, Cys168 in the active site of DIO3 is believed to act as the resolving cysteine. Given this, it is unlikely that the chalcogen bond interaction between the -SH and -SeH groups fixed at the 1,8-positions of the naphthalene ring in mimic 1 accurately reflects the interaction between Cys168 and Sec170 in the enzyme. Therefore, the phrase “closely mimicking the primary structure of DIO3” seems inappropriate.

  The authors themselves note in lines 393–397:  

  “While DIO1 and DIO3 both contain conserved cysteine and selenocysteine residues in their active sites, the naphthyl-based mimic having a thiol and selenol in close proximity to each other was unable to remove phenolic ring iodine from T4. This suggests that other factors are likely to influence the reactivity and regioselectivity of deiodination of T4 by mimic.”

  It would improve the clarity of the discussion if the authors underscore that the activation via chalcogen bond interactions is a phenomenon specific to the synthetic mimic and does not necessarily reflect the enzymatic mechanism. It is indeed intriguing that such chalcogen bond interactions enabled deiodination of T4 under physiological conditions in bulk solution, where activation and conformational optimization by an active site, as seen in enzymatic reactions, are absent. However, this distinction should be made clear.

2. Before the sentence beginning “This loss is attributed…” on line 232, it appears that the discussion of the Cys derivative (Compound 11) is missing. Please include the relevant information or discussion regarding this compound.

3. In Figure 7A, the central structure should show “OH” instead of “OR.”

Author Response

Thank you very much for taking the time to review this manuscript. Please find below the detailed responses along with the corresponding revisions and corrections, which are highlighted in the revised manuscript.

Reviewer comment 1: On lines 175–177, the authors state,

Due to the rigidity of the aromatic ring, the selenol and thiol moieties are held close to each other, closely mimicking the primary structure of DIO3.”

However, as noted in lines 276–278, Cys168 in the active site of DIO3 is believed to act as the resolving cysteine. Given this, it is unlikely that the chalcogen bond interaction between the -SH and -SeH groups fixed at the 1,8-positions of the naphthalene ring in mimic 1 accurately reflects the interaction between Cys168 and Sec170 in the enzyme. Therefore, the phrase “closely mimicking the primary structure of DIO3” seems inappropriate.

Our response: We agree with the reviewer that the phrase "closely mimicking the primary structure of DIO3" is inappropriate. In the active enzyme, the thiol (Cys 168) and selenol (Sec 170) groups are positioned at a greater distance, than as observed in the mimic (3 Å). This suggests that, while the mimic exhibits activity similar to DIO3, it differs in its actual structure. Due to the rigidity of the aromatic ring, it holds the thiol and selenol groups closer to each other. Therefore, we have revised our statement in the manuscript and removed the phrase "closely mimicking the primary structure of DIO3 (Line 173-176)”.

The authors themselves note in lines 393–397

“While DIO1 and DIO3 both contain conserved cysteine and selenocysteine residues in their active sites, the naphthyl-based mimic having a thiol and selenol in close proximity to each other was unable to remove phenolic ring iodine from T4. This suggests that other factors are likely to influence the reactivity and regioselectivity of deiodination of T4 by mimic.”

It would improve the clarity of the discussion if the authors underscore that the activation via chalcogen bond interactions is a phenomenon specific to the synthetic mimic and does not necessarily reflect the enzymatic mechanism. It is indeed intriguing that such chalcogen bond interactions enabled deiodination of T4 under physiological conditions in bulk solution, where activation and conformational optimization by an active site, as seen in enzymatic reactions, are absent. However, this distinction should be made clear.

Our response: We have now clearly explained that the deiodination observed in the mimics differs from that in DIOs. In the case of DIOs, other factors, such as substrate binding to the active site, alterations in the active site conformation, and the conformational changes of the substrate, play a key role in the deiodination process. These factors are absent in the deiodination reaction by mimics. We have thoroughly discussed these differences in reactivity in the revised manuscript (Line 415-424).

Reviewer comment 2: Before the sentence beginning “This loss is attributed…” on line 232, it appears that the discussion of the Cys derivative (Compound 11) is missing. Please include the relevant information or discussion regarding this compound.

Our response: We apologize for this mistake. In the revised manuscript, we have now included a discussion about compound 11 having a seleno-sulfide moiety (Line 231-238).

Reviewer comment 3: In Figure 7A, the central structure should show “OH” instead of “OR.”

Our response: We have changed the OR group to the OH group and included the correct figure in the revised manuscript (Figure 7A).